# Mechanisms of the RNA helicases DDX42 and DDX46 in human U2 snRNP assembly

Fenghua Yang[1,2,3,6], Tong Bian [1,2,3,6], Xiechao Zhan [2,3,6], Zhe Chen [4], Zhihan Xing[2,3], Nicolas A. Larsen [5], Xiaofeng Zhang [2,3] ✉ & Yigong Shi [2,3] ✉

Three RNA helicases – DDX42, DDX46 and DHX15 – are found to be associated with human U2 snRNP, but their roles and mechanisms in U2 snRNP and spliceosome assembly are insufficiently understood. Here we report the cryo-electron microscopy (cryo-EM) structures of the DDX42-SF3b complex and a putative assembly precursor of 17S U2 snRNP that contains DDX42 (DDX42-U2 complex). DDX42 is anchored on SF3B1 through N-terminal sequences, with its N-plug occupying the RNA path of SF3B1. The binding mode of DDX42 to SF3B1 is in striking analogy to that of DDX46. In the DDX42-U2 complex, the N-terminus of DDX42 remains anchored on SF3B1, but the helicase domain has been displaced by U2 snRNA and TAT-SF1. Through in vitro assays, we show DDX42 and DDX46 are mutually exclusive in terms of binding to SF3b. Cancer-driving mutations of SF3B1 target the residues in the RNA path that directly interact with DDX42 and DDX46. These findings reveal the distinct roles of DDX42 and DDX46 in assembly of 17S U2 snRNP and provide insights into the mechanisms of SF3B1 cancer mutations.

Splicing of pre-mRNA is executed by an exceptionally dynamic RNA-protein complex known as the spliceosome[1-3]. The spliceosome is assembled from five small nuclear ribonucleoprotein particles (snRNPs), known as U1, U2, U4, U5 and U6, and non-snRNP factors[4]. Each snRNP is built with an individual small nuclear RNA (snRNA), seven common Sm proteins (or LSm in the case of U6), and a number of specific protein factors[5]. Among the five snRNPs, the U2 snRNP plays essential roles during the recognition of the intron and assembly of prespliceosome[6-10].

The human U2 snRNP is especially complex and dynamic; its assembly is a multi-step and poorly understood process[11-13]. The 17S U2 is regarded as the functional form of human U2 snRNP that directly participates in early spliceosome assembly[11]. The core components of 17S U2 snRNP comprises the SF3b complex, SF3a complex, 12S U2 core

(U2 snRNA, Sm ring, U2-A' and U2-B"), the splicing factor TAT-SF1 and DDX46[11-18]. In vitro assays have suggested a sequential assembly of SF3b with 12S U2 core, forming an intermediate known as 15S particle, followed by SF3a complex[11-13,18]. However, little is known about the protein factors that orchestrate this assembly process.

Notably, three RNA-dependent ATPase – DDX42, DDX46 and DHX15 – are found to be associated with U2 snRNP[11,19]. The first two are members of the DEAD-box helicase family while DHX15 belongs to DEAH-box subfamily. DDX42 co-purifies with SF3b complex but is nearly undetectable in 17S U2 snRNP[11]. Therefore, DDX42, which has RNA chaperone activity[20], is likely released after fulfilling its role in the assembly of U2 snRNP[11]. In contrast to DDX42, DDX46 (yeast Prp5 homolog) is an integral component of 17S U2 and plays essential roles during assembly of prespliceosome and proofreading of the branch

[1]College of Life Sciences, Fudan University, Shanghai 200433, China. [2]Research Center for Industries of the Future, Key Laboratory of Structural Biology of Zhejiang Province, School of Life Sciences, Westlake University; Institute of Biology, Westlake Institute for Advanced Study, 18 Shilongshan Road, Hangzhou 310024 Zhejiang Province, China. [3]Westlake Laboratory of Life Sciences and Biomedicine, 18 Shilongshan Road, Hangzhou 310024 Zhejiang Province, China. [4]State Key Laboratory of Pathogenesis, Prevention and Treatment of High Incidence Diseases in Central Asia, Xinjiang Key Laboratory of Molecular Biology for Endemic Diseases, Department of Biochemistry and Molecular Biology, School of Basic Medical Sciences, Xinjiang Medical University, Urumqi 830017 Xinjiang, China. [5]Foghorn Therapeutics, 500 Technology Square, Suite 700, Cambridge, MA 02139, USA. [6]These authors contributed equally: Fenghua Yang, Tong Bian, Xiechao Zhan. ✉e-mail: xiaofengzhang@ustc.edu.cn; syg@westlake.edu.cn

site[6–10,21–24]. But whether DDX46 has additional roles in U2 snRNP assembly is unclear. In addition, while the mechanism of DHX15 (yeast Prp43 ortholog) in disassembly of intron-lariat spliceosome (ILS) complex has been extensively investigated[25–28], its roles in U2 snRNP and early splicing complex remain enigmatic[29].

Addressing these key questions necessitates structural and functional elucidation of these RNA helicases. Recent studies have revealed both the overall architecture of 17S U2 snRNP[22–24], and the detailed interactions between SF3B1 and DDX46[23]. However, structural information on DDX42 and DHX15 in the early splicing complex is still lacking. In this manuscript, we report the high-resolution structures of the DDX42-SF3b complex and a putative assembly intermediate of 17S U2 snRNP that contains DDX42 (hereafter referred to as DDX42-U2 complex). We also isolated a form of U2 snRNP that contains DHX15, but the position of DHX15 was not identified by the EM density map. Our structures reveal a shared striking pattern of SF3B1 interactions with DDX42, DDX46, and the polypyrimidine-tract (PPT) of pre-mRNA. Together with structure-guided biochemical analysis, our study reveals a coherent picture on the roles of DDX42 and DDX46 in U2 snRNP assembly and provide insights into SF3B1 cancer mutations.

## Results

### Sample preparation and structure determination

The RNA helicases DDX42 and DDX46 were detected stoichiometrically in the immunoprecipitates of SF3B1, a central component of U2 snRNP (Fig. 1a). Peptides derived from DHX15 were also identified, but in lower abundance (Fig. 1a). Structural information on these RNA helicases is key to understanding the assembly of 17S U2 snRNP and prespliceosome. First, we sought to purify the DDX42-SF3b complex. DDX42 was co-expressed with four protein components of the SF3b complex (SF3B1, SF3B3, SF3B5 and PHF5A) in HEK293F cells and purified using tandem affinity chromatography. The purified complex was further fractionated on gel filtration (Supplementary Fig. 1a). The peak fractions were imaged using a Titan Krios electron microscope equipped with a K3 detector (Supplementary Fig. 1b, c). After two-dimensional (2D) and three-dimensional (3D) classifications, 234,800 particles result in a reconstruction of the DDX42-SF3b complex at an average resolution of 2.6 Å (Fig. 1b, Supplementary Fig. 2, Supplementary Tables 1, 2). Soft masks were applied to the N-terminus and the helicase domain of DDX42, which reside in the peripheral regions of the complex. These measurements improved the local resolutions (Supplementary Fig. 1c).

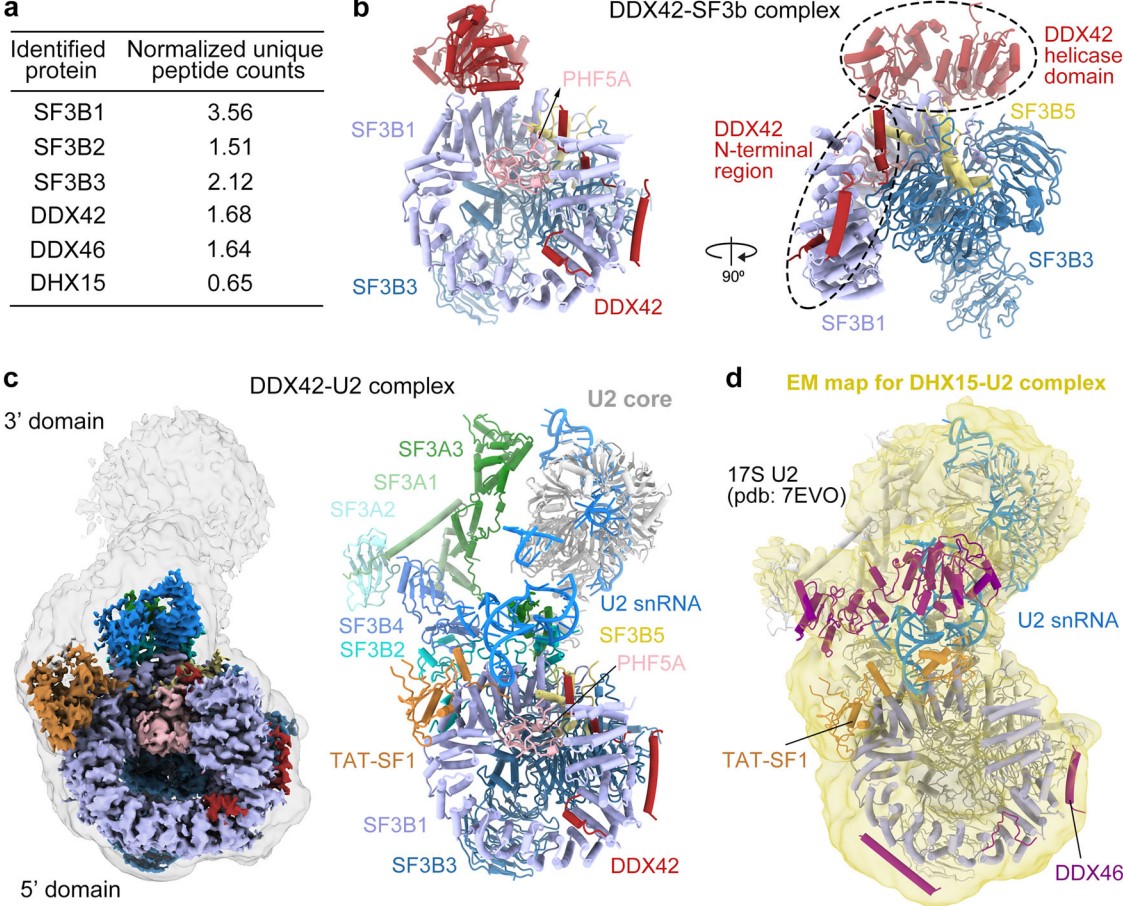

**Fig. 1 | Cryo-EM structures of the DDX42-SF3b complex and the human DDX42-U2 complex. a** Mass spectrometric analyses identified peptides derived from DDX42, DDX46 and DHX15 in immunoprecipitates prepared from HEK293F cells that expressed Flag-tagged SF3B1. DDX42 and DDX46 are detected stoichiometrically with SF3B1. In contrast, the peptides of DHX15 are less abundant. **b** Overall structure of the DDX42-SF3b complex. The SF3b core consists of four components: SF3B1, SF3B3, SF3B5 and PHF5A. The DEAD-box ATPase/helicase DDX42 (colored in red) is bound at the periphery of the complex. **c** The cryo-EM map for the DDX42-U2 complex has a bi-lobal shape. The high-resolution map of the 5' domain (colored by chain identity) is embedded in a low-pass filtered map that showing the position of the 3' domain (left panel). The 3' domain was modeled by docking of the previously reported coordinates (pdb: 7EVO)[23]. The final model includes the SF3b complex, SF3a complex, U2 core (Sm ring, U2-A', U2-B'' and U2 snRNA), the splicing factor TAT-SF1 and the N-terminal region of DDX42 (right panel). **d** Docking of the atomic model of 17S U2 snRNP[23] into the low-pass filtered EM density map for the DHX15-U2 complex (colored in yellow). DDX46 (colored in purple) and TAT-SF1 (colored in orange) are present in DHX15-U2 complex; DHX15 could not be identified by the EM density map.

Next, to purify potential spliceosomal complex that contains DDX42, a Flag tag was fused to the N-terminus of DDX42 and was employed for affinity selection of endogenous complexes. The purified sample was applied to glycerol gradient centrifugation under crosslinking condition[30]. The final sample contains U2 only of the five snRNAs, DDX42 and protein subunits of U2 snRNP (DDX42-U2 complex) (Supplementary Fig. 3a, Supplementary Table 3). Five thousand and thirty-nine micrographs were collected, generating 2,698,248 particles (Supplementary Fig. 3b). After 2D and 3D classifications, 430,443 particles yielded a final reconstruction of human DDX42-U2 complex at an average resolution of 2.7 Å (Fig. 1c, Supplementary Fig. 3c–f, Supplementary Tables 1, 2).

Using a similar strategy, we employed a Flag-tag on DHX15 to pulldown endogenous splicing complexes. After affinity selection and glycerol gradient centrifugation, we isolated a macro-molecular complex that contains DHX15, U2 snRNA and most protein components of 17S U2 snRNP (referred to as DHX15-U2 complex) (Supplementary Fig. 4a, b). Cryo-EM analysis yielded a reconstruction for DHX15-U2 complex at a moderate resolution of about 5 Å (Supplementary Fig. 4c–e). The EM map displays similar structural features comparing to that of 17S U2 snRNP. Indeed, the entire atomic model of 17S U2 snRNP[23] could be readily fitted into the EM density map (Fig. 1d, Supplementary Fig. 4f). However, the RNA helicase DHX15 could not be identified, likely due to the potential flexibility between DHX15 and U2 snRNP. Thus, we will focus on DDX42 and DDX46 in the following sections of this manuscript.

### Anchoring of DDX42 on the SF3b complex

In the DDX42-SF3b complex, SF3B1, SF3B3, SF3B5 and PHF5A adopt a nearly identical conformation as that of the free SF3b complex[31,32]. DDX42 is bound at the periphery of the SF3b core (Fig. 1b) and is anchored on the core mainly through its N-terminal extended sequences (Fig. 2a, Supplementary Fig. 5). These sequences of DDX42 constitute three discrete motifs: an N-terminal plug (N-plug, residues 69–81), an extended helix $\alpha_{-3}$ (residues 114–135), and a pair of short helices $\alpha_{-2}/\alpha_{-1}$ (residues 145–167) (Fig. 2b). The N-plug and helix $\alpha_{-3}$ are anchored on the HEAT repeats (HRs) of SF3B1 (SF3B1$^{HEAT}$), a scaffolding component of the SF3b complex (Fig. 2c). The helices $\alpha_{-2}/\alpha_{-1}$ are bound in a surface cleft at the junction among SF3B1, SF3B3 and SF3B5 (Supplementary Fig. 6a). In contrast to these sequence motifs, the bulky helicase domain (residues 207–643) of DDX42, which consists of two RecA domains, is loosely attached to the C-terminus of SF3B1$^{HEAT}$; this interface is characterized by weak cryo-EM density (Fig. 2a, b; Supplementary Fig. 5g).

The most notable feature of the DDX42-SF3b interface is the placement of the DDX42 N-plug in a surface groove of SF3B1 (Fig. 2d, Supplementary Fig. 6b). This positively charged groove is exactly the RNA path where the polypyrimidine tract (PPT) of pre-mRNA binds[33–35] (Supplementary Fig. 6c). In our structure, the N-plug is lodged in the RNA path, with the negatively charged amino acids from the N-plug making a network of hydrogen bonds (H-bonds) with the positively charged residues from HR4-HR9 (Fig. 2d, Supplementary Fig. 6b). In particular, the acidic carboxylates of Glu71, Glu72, Glu77, Asp78, and Glu80 from DDX42 accept charge-stabilized H-bonds from the basic side chains of Arg828, Lys748, His662/Lys666/Lys700, Lys741, and Arg625/Lys700 from SF3B1, respectively. The N-plug sequences are highly conserved from zebrafish to human, suggesting a conserved pattern of interaction (Supplementary Fig. 6b).

In contrast to the N-plug, the $\alpha_{-3}$ helix of DDX42 binds the outside of HR3 through HR5 of SF3B1 (Fig. 2e). Asp114/Glu117 and Asp130 at the N- and C-termini of helix $\alpha_{-3}$ make H-bonds to Lys656 and Arg568 of SF3B1, respectively. A stripe of four hydrophobic amino acids Ile116/Phe119/Met120/Val123 on helix $\alpha_{-3}$ make van der Waals contacts to a greasy surface epitope formed by Val651/Phe647/Ile610 of SF3B1.

Similar to the N-plug, the short helices $\alpha_{-2}/\alpha_{-1}$ are enriched by negatively charged amino acids, which are recognized by positively charged residues in a surface cleft formed by SF3B1, SF3B3, SF3B5, and PHF5A (Fig. 2f, Supplementary Fig. 6d).

Compared to the well-defined interfaces mediated by the three N-terminal motifs of DDX42, the interface between its helicase domain and the SF3b core is characterized by weak EM density (Supplementary Fig. 5g), suggesting weak association. This structural organization is likely required by the activity of an RNA helicase that depends on ATP binding and hydrolysis: translocation of single-stranded RNA necessitates considerable leeway for the helicase domain.

### Overall structure of the DDX42-U2 complex

The overall architecture of DDX42-U2 complex is similar to that of 17S U2 snRNP (pdb: 7EVO)[23] (Fig. 3a); both of them exhibit a bipartite organization, with the 3′ domain flexibly connected to the 5′ domain (Fig. 1c). The resolution of cryo-EM map for the 5′ domain reaches to 2.7 Å, allowing identification of detailed structural features (Fig. 1c, Supplementary Figs. 3 and 7). The 3′ domain is less-well resolved, but we were able to perform rigid-body docking of known structures into the EM map. Compared to 17S U2, the conformations of SF3b, SF3a and the U2 snRNA in the DDX42-U2 complex are nearly identical (Fig. 3a). For example, the branchpoint recognition sequence (BPRS) of U2 snRNA forms a structure known as branchpoint-interacting stem-loop (BSL)[10] (Fig. 3a); its conformation in DDX42-U2 complex remains largely unchanged in 17S U2 snRNP[10,36]. The most significant difference is the presence of DDX42, rather than DDX46, in the DDX42-U2 complex (Fig. 3a). In addition, the UHM domain of TAT-SF1, which together with the Linker domain sandwiches BSL of the U2 snRNA, is missing in the structure of DDX42-U2 complex (Fig. 3a, b; Supplementary Fig. 7b).

### Displacement of the helicase domain of DDX42

DDX42 copurifies with SF3b complex, but is released upon formation of 17S U2 snRNP[11]. Our DDX42-U2 complex likely represents an intermediate state in which DDX42 has fulfilled its function but is yet to be released. Compared to DDX42-SF3b complex, the N-terminal sequences of DDX42, including the N-plug and helices $\alpha_{-3}/\alpha_{-2}/\alpha_{-1}$, remain bound to SF3B1, but the helicase domain has been displaced from the C-terminus of SF3B1$^{HEAT}$ (Figs. 2a and 3b). A continuous EM map lobe that lies on top of BSL was observed, but the local resolution does not allow unambiguous structural docking (Supplementary Fig. 7c). Thus the exact location of DDX42 helicase domain in the DDX42-U2 complex remains unclear. In addition, splicing factor TAT-SF1 and BSL of the U2 snRNA are recruited to SF3B1$^{HEAT}$, anchoring on 15–18 and 18–20 HEAT repeats, respectively (Fig. 3b).

Conformation of U2 snRNA is especially dynamic[37,38]. The nucleotides of BPRS and part of the stem-loop I (SL I) of U2 snRNA are remodeled to form BSL during the formation of 17S U2 snRNP[10,36]. Structural overlay of DDX42-SF3b with DDX42-U2 reveals overlapping positions between the helicase domain of DDX42 and BSL of the U2 snRNA (Fig. 3b inset). This analysis suggests a potential role of DDX42 in the formation of BSL. In addition, the helicase domain of DDX42 also overlaps with the Linker domain of TAT-SF1 (Fig. 3b inset), which contributes to maintaining the BSL conformation[10,39–41]. Our 2.7 Å reconstruction of DDX42-U2 complex allows unambiguous identification of two interfaces between TAT-SF1 and SF3B1. The Linker domain of TAT-SF1 mainly interacts with residues from HR16 and HR17 of SF3B1$^{HEAT}$. This interface features a charge-stabilized H-bond and a cation-π interaction between Asp252/Trp249 from TAT-SF1 and Arg1106/Arg1109 from SF3B1, respectively (Fig. 3c). In contrast, the RRM domain of TAT-SF1 mainly interacts with HR15 and HR16 through a combination of H-bonds and van der Waals contacts (Fig. 3d). These interactions likely play crucial roles in displacing the helicase domain of DDX42 from SF3B1.

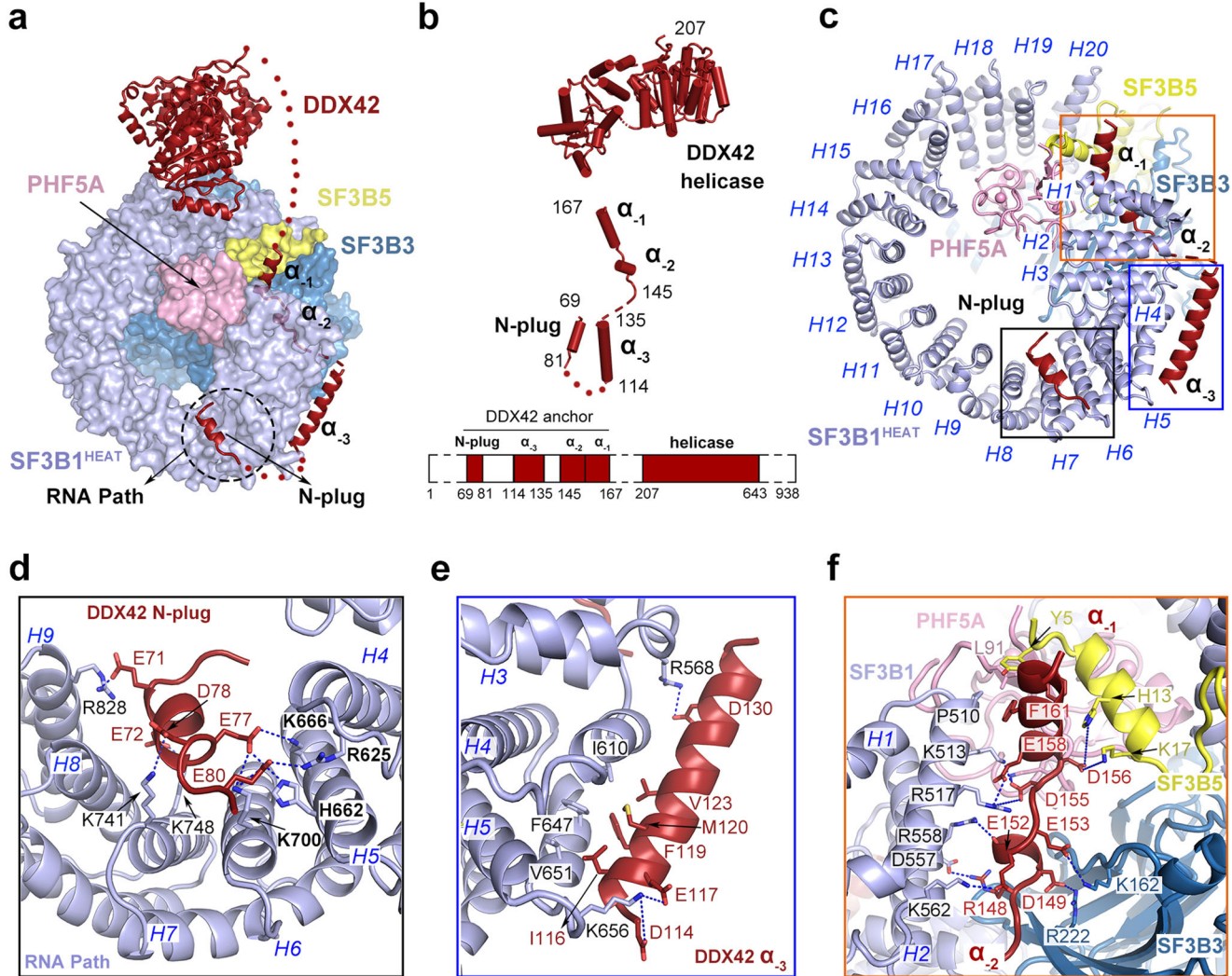

**Fig. 2 | The RNA helicase DDX42 is anchored on SF3b through three distinct interfaces. a** DDX42 is anchored on the SF3b complex through three distinct interfaces mediated by the N-plug and the helices $\alpha_{-3}$, $\alpha_{-2}$, and $\alpha_{-1}$. The SF3b core is shown in surface view and DDX42 is displayed in cartoon representation. The N-plug of DDX42 occupies the RNA path of SF3B1. **b** The structure of DDX42 comprises an N-terminal plug (N-plug), three helices $\alpha_{-3}/\alpha_{-2}/\alpha_{-1}$, and a C-terminal helicase domain. A schematic diagram of the DDX42 domain organization is shown below the cartoon. **c** Three N-terminal motifs of DDX42 mainly interact with SF3B1$^{HEAT}$. The N-plug and the helix $\alpha_{-3}$ bind to HEAT repeats 4–9 and 3–5, respectively. The $\alpha_{-2}/\alpha_{-1}$ helices are bound at the junction among the four components of the SF3b core. Each HEAT repeat is indicated by italic H followed by its repeat number. **d** A close-up view on the N-plug of DDX42 and its interactions with residues from HEAT repeats 4–9 of SF3B1. Negatively charged residues from the N-plug form specific H-bonds with positively charged amino acids in the RNA path of SF3B1. The SF3B1 residues targeted by cancer-associated hotspot mutations, such as Lys700, are highlighted in bold. **e** The elongated helix $\alpha_{-3}$ associates with the lateral side of HEAT repeats 3–5 of SF3B1 through a combination of H-bonds and van der Waals contacts. **f** The $\alpha_{-2}/\alpha_{-1}$ helices are lodged in a surface cavity at the junction formed by SF3B1, SF3B3, SF3B5 and PHF5A. This cavity is enriched by positively charged residues, which make a number of specific H-bonds to negatively charged amino acids in DDX42.

## Sequential engagement of the RNA path

Structural comparison of DDX42-SF3b and DDX42-U2 with previously reported 17S U2[23] and assembled spliceosome[33] shows that the RNA path of SF3B1 is sequentially occupied by three structural motifs: the N-plug of DDX42 in DDX42-SF3b and DDX42-U2, the acidic loop of DDX46 in 17S U2, and the polypyrimidine tract (PPT) of pre-mRNA in assembled spliceosome (Fig. 4a). Intriguingly, both the N-plug (Supplementary Fig. 6b), the acidic loop[23] and the PPT are highly negatively charged and their interactions with the RNA path are mutually exclusive (Fig. 4b). Superimposition of SF3B1 from DDX42-U2, 17S U2 and B$^{act}$ spliceosome reveals the overlapping binding sites in the RNA path by aforementioned three motifs (Fig. 4b inset). In addition, the N-terminal sequences of DDX42 and DDX46 share additional anchoring sites on the convex surface of SF3B1. For example, the helix $\alpha_{-2}$ from DDX42 and helix $\alpha2$ from DDX46 bind SF3B1 at similar positions

(Fig. 4b). Thus, release of DDX42 from SF3B1 is a prerequisite for DDX46 recruitment during maturation of 17S U2 snRNP, explaining previous biochemical observation[11].

Given the RNA path is highly positively charged and may be accessible to non-specific RNA and other proteins (Supplementary Fig. 6b, c); its occupation by the N-plug of DDX42 or the acidic loop of DDX46 effectively prevents such unproductive binding. In this sense, both DDX42 and DDX46 serve as chaperones of the RNA path and preserve its function of engaging the next interacting partner.

## DDX42 and DDX46 bind SF3b in a competitive manner

To facilitate mechanistic understanding of DDX42 and DDX46, we quantitatively measured their interactions with the SF3b complex using bio-layer interferometry. DDX42 and DDX46 display apparent binding affinities of 23 nM and 66 nM towards the SF3b complex,

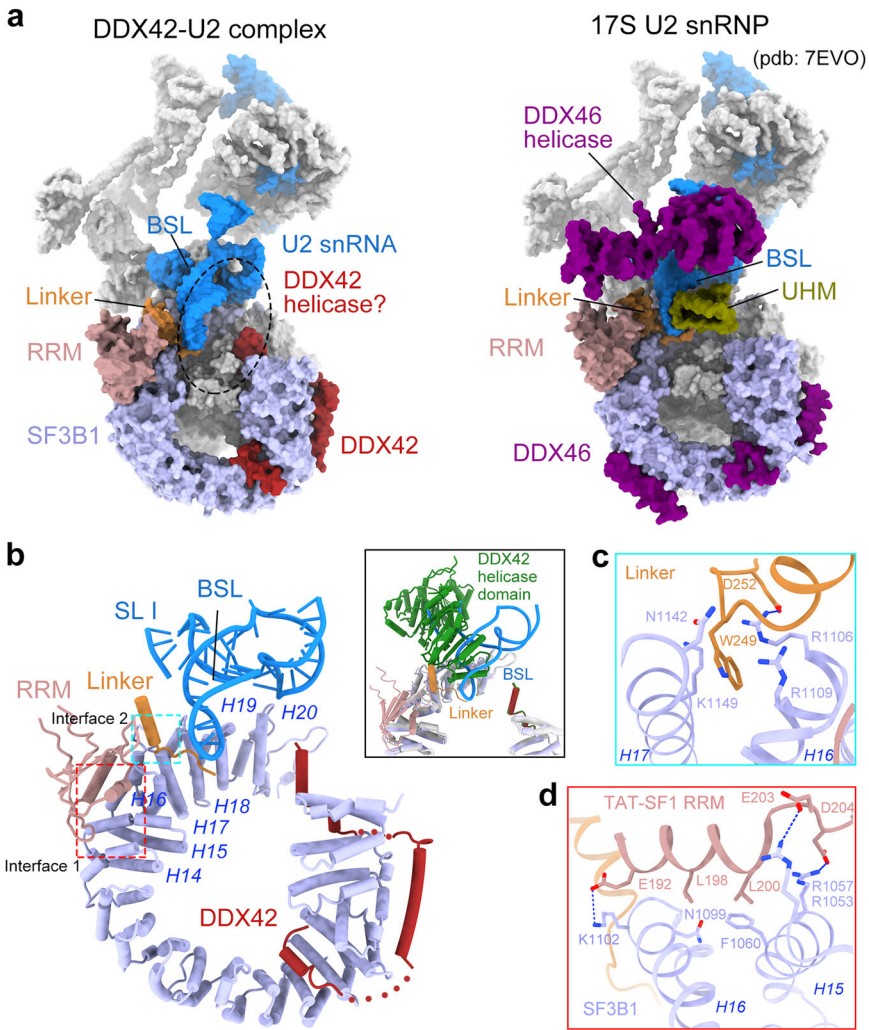

**Fig. 3 | Similarities and differences between the structures of DDX42-U2 complex and 17S U2 snRNP. a** Surface representation of the DDX42-U2 complex (left panel) and 17S U2 snRNP (right panel) (pdb: 7EVO)[23] models. In DDX42-U2, the N-terminus of DDX42 remains anchored on SF3B1, but the helicase domain has been displaced comparing to that in DDX42-SF3b complex. The potential location of the helicase domain of DDX42 is indicated by a dashed circle. In 17S U2, the RNA helicase DDX46 replaces DDX42 to interact with SF3B1. Positions of the RRM, Linker and UHM domains of TAT-SF1 are indicated. The UHM domain, which is associated with U2 snRNA in 17S U2 snRNP, is missing in the structure of DDX42-U2. The conformations of the branchpoint-interacting stem-loop (BSL) are identical among DDX42-U2 and 17S U2 snRNP. **b** The BSL and TAT-SF1 are anchored on the C-terminal HEAT repeats of SF3B1; this position overlaps with the anchoring site of DDX42 helicase domain. The inset shows the superimposition of the structures of the DDX42-SF3b complex and DDX42-U2 complex. Protein components and U2 snRNA of the DDX42-U2 complex are color coded. DDX42 and SF3B1 from the DDX42-SF3b complex are shown in green and grey, respectively. Two interfaces between TAT-SF1 and SF3B1 were identified by our high-resolution reconstruction. **c** A close-up view on the interface between the Linker domain of TAT-SF1 and SF3B1. **d** A close-up view on the interface between the RRM domain of TAT-SF1 and SF3B1.

respectively (Fig. 4c). Because both DDX42 and DDX46 occupy the RNA path of SF3B1, they are predicted to be competitive in terms of binding to the SF3b. This is indeed the case. When supplied in large excess over DDX46, DDX42 displaced DDX46 from the pre-assembled DDX46-SF3b complex to form the DDX42-SF3b complex (Fig. 4d, lanes 3,6,7). Similarly, DDX46 in large excess was able to displace DDX42 from the DDX42-SF3b complex (Fig. 4d, lanes 2,4,5). This competitive binding may facilitate the maturation of 17S U2 snRNP.

## Discussion

In our working model, DDX42 is spontaneously associated with SF3b, forming the DDX42-SF3b complex. Then the 12S U2 core particle and SF3a are recruited to assemble the DDX42-U2 complex. Although the target of DDX42 activity remains unclear, our structures suggest DDX42 may play essential roles during the accommodation of U2 snRNA on SF3B1 (Figs. 3b and 5a). Stable accommodation of BSL on SF3B1$^{HEAT}$ results in the displacement of the helicase domain of DDX42,

which in turn allows the recruitment of TAT-SF1. In DDX42-U2, the Linker domain stabilizes the BSL conformation, but the UHM domain is relatively mobile (Fig. 3a, Supplementary Fig. 7b). Displacement of the helicase domain, together with the competition from the N-terminus of DDX46, eventually resulting in dissociation of DDX42 (Fig. 4d). During this process, the RNA path is handed over from the N-plug of DDX42 to the acidic loop of DDX46 (Fig. 4a). Loading of DDX46 further stabilizes the UHM domain of TAT-SF1 through protein-protein interactions as suggested by previous crosslinking data[36]. Therefore, the replacement of DDX42 by DDX46 may constitute a turning point in the maturation process of 17S U2 snRNP.

Somatic mutations in SF3B1 lead to dysregulated RNA splicing and are observed in a variety of cancers[42–44]. A large majority of these mutations were shown to abrogate the binding of DDX42 and DDX46[23,45,46]. The SF3B1 residues that interact with DDX42 or DDX46 are grouped into three classes: with DDX42 only, with DDX46 only, and with both (Supplementary Table 4). Intriguingly, the SF3B1 residues

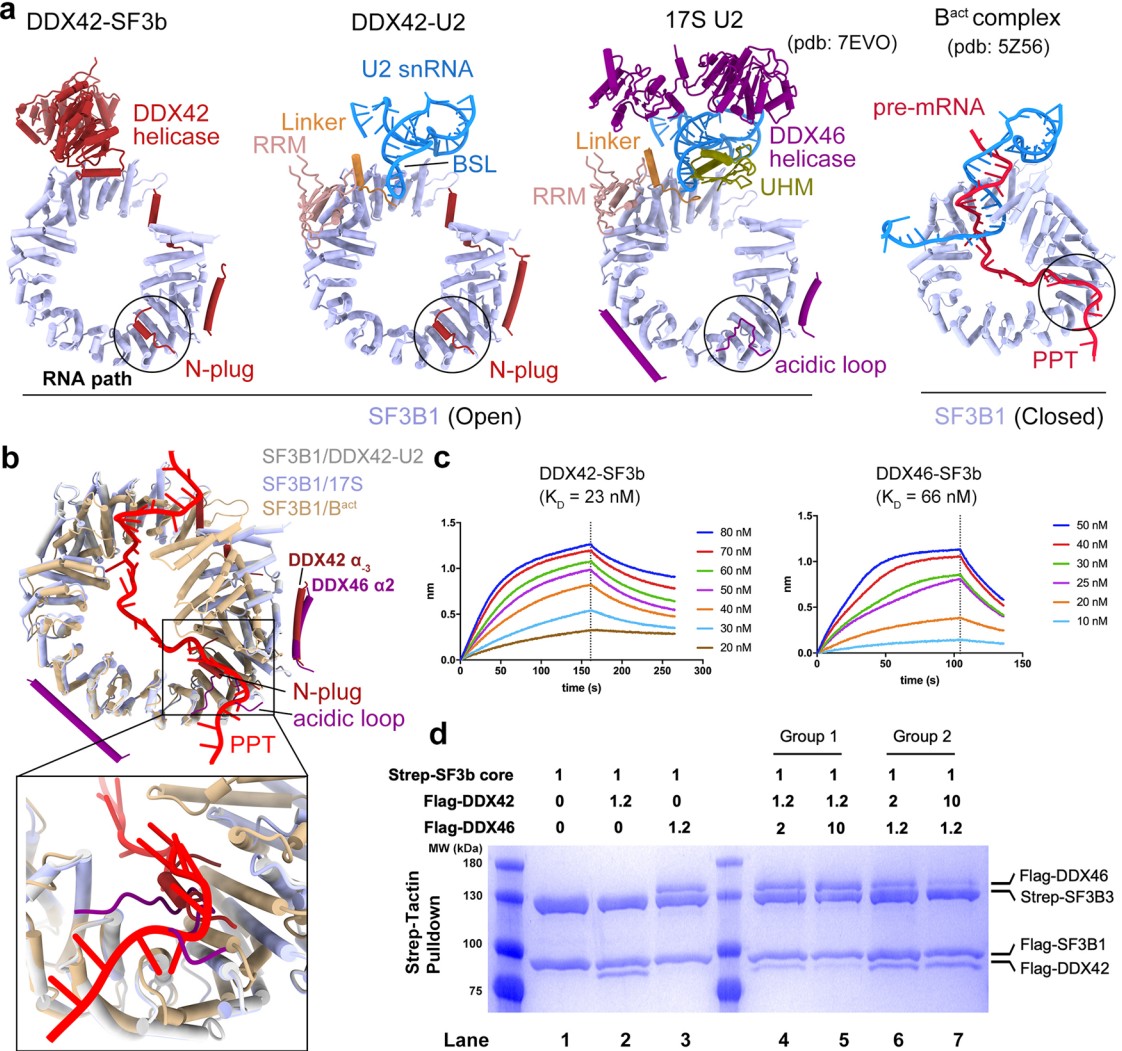

**Fig. 4 | The RNA path is sequentially engaged by the N-plug of DDX42, the acidic loop of DDX46 and pre-mRNA. a** The RNA path of SF3B1 is sequentially occupied by three structural motifs: the N-plug of DDX42 in DDX42-SF3b and DDX42-U2, the acidic loop of DDX46 in 17S U2, and the polypyrimidine tract (PPT) of pre-mRNA in assembled spliceosome. Shown here is a structural comparison among the relevant components from the DDX42-SF3b complex (left panel), DDX42-U2 complex (middle-left panel), 17S U2 snRNP[23] (middle-right panel), and the B[act] complex[33]. The RNA path is identified by black circles. **b** DDX42, DDX46 and PPT are mutually exclusive in terms of binding to the RNA path of SF3B1. Shown here is the overlay of relevant structural elements from DDX42-U2, 17S U2 and B[act]

complex. The alignment is made through SF3B1. **c** DDX42 binds to the SF3b with an apparent dissociation constant of 23 nM. Shown here are results of the bio-layer interferometry for measurement of the binding kinetics between the SF3b and DDX42 (left panel). Similarly, DDX46 binds to the SF3b with an apparent dissociation constant of 66 nM (right panel). **d** DDX42 and DDX46 bind SF3b in a competitive manner. Shown here are results of competition assay between DDX42 and DDX46 for binding to the SF3b core complex. DDX46 in large excess over DDX42 is able to replace DDX42 in the pre-assembled DDX42-SF3b complex, and vice versa. The assays were independently repeated for three times with similar results.

that interact with DDX42 only are spared by cancer-derived mutations. In contrast to DDX42, three SF3B1 residues (Asp894, Tyr898 and Glu902) that interact with DDX46 only are mutated in cancer. In addition, seven SF3B1 residues that interact with both DDX46 and DDX42 are targeted for mutations in cancer, including the most frequently mutated residue Lys700. Remarkably, all seven residues are located in the RNA path and directly recognize the N-plug of DDX42 and the acidic loop of DDX46 (Figs. 2d and 5b); substitutions of these residues in cancers impair the binding of both DDX42[46] and DDX46[23].

Our study suggests distinct mechanisms of DDX42 and DDX46 in cancer development. The crippled DDX46 binding to U2 snRNP may lead to loss of branch site proofreading function by DDX46[23,47–50]. In contrast, the crippled DDX42 binding to SF3B1 may lead to inappropriate assembly of 17S U2 snRNP (Figs. 4 and 5a). In either scenario, assembly of prespliceosome could be damaged. In addition, the compromised binding of DDX42 and DDX46 to SF3B1 may lead to

premature exposure of the RNA path to pre-mRNA and other protein factors, which may harm the function of U2 snRNP. This analysis identifies important roles of DDX42 N-plug and DDX46 acidic loop in mediating the effects of cancer mutations in SF3B1.

Apart from DDX42 and DDX46, the DEAH-box helicase DHX15 is present in purified U2 snRNP, along with several G-patch proteins exemplified by TFIP11 and CHERP[11,19] (Supplementary Fig. 4b). Although our EM map fails to reveal the critical interactions between DHX15 and U2 snRNP, docking of 17S U2 snRNP into the EM map of the DHX15-U2 complex suggests DHX15 is not a core component of U2 snRNP, and may bind at the peripheral region of U2 snRNP either directly or through its G-patch co-factors (Fig. 1d, Supplementary Fig. 4f). It is also unlikely that DHX15 is involved in facilitating 17S U2 snRNP assembly. Notably, by taking advantage of differential nucleotide triphosphates requirements of DEAD and DEAH helicases, a recent study suggests a quality control function of DHX15 during

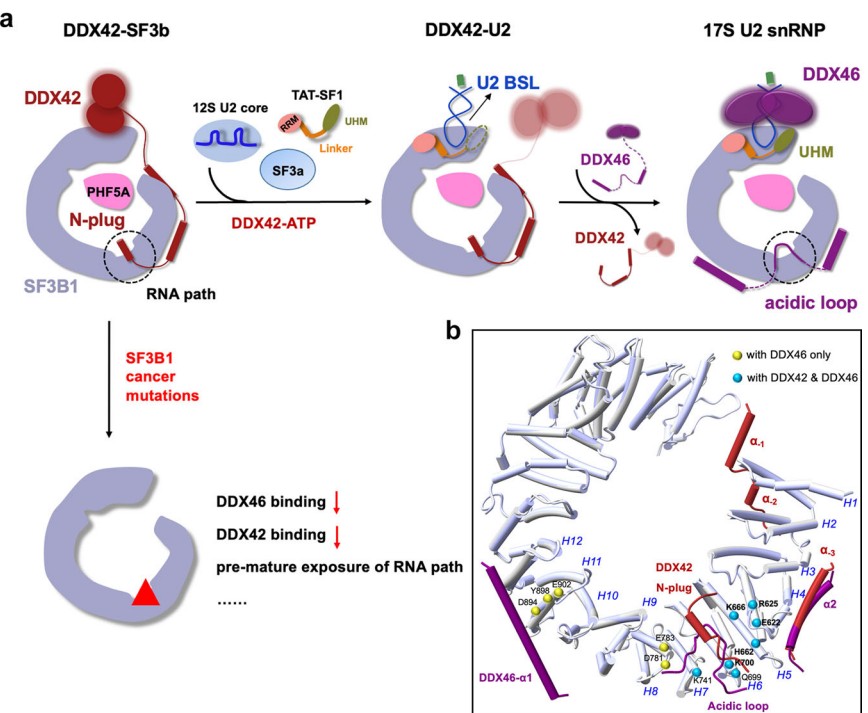

**Fig. 5 | A working model for DDX42 and DDX46 in U2 snRNP assembly and SF3B1 cancer mutations. a** A carton diagram of the working model on 17S U2 snRNP formation. In the default state, the SF3b core forms a stable complex with DDX42, where the RNA path of SF3B1 is occupied by the N-plug of DDX42. In step 1, DDX42 associates with U2 snRNA from 12S U2 core and likely drives its conformational change towards formation of BSL. During this process, the SF3a complex, TAT-SF1 are recruited into the complex, resulting in the displacement of DDX42 helicase domain and formation of DDX42-U2 complex. The UHM domain of TAT-SF1 in DDX42-U2 is relatively mobile and thus indicated as a dashed circle. In step 2, the movement of DDX42 helicase domain might destabilized the interaction between its N-terminus and SF3B1; this may facilitate DDX46 to outcompete DDX42 from SF3B1, forming mature 17S U2 snRNP. The UHM domain of TAT-SF1, which interacts with DDX46[36], may help stabilize the conformation of BSL and the helicase domain of DDX46. **b** The vast majority of cancer-derived mutations in SF3B1 are clustered at its interface with DDX42 and DDX46. Key SF3B1 cancer-related residues that interact with DDX46 only and both DDX42 and DDX46 are indicated by yellow and cyan spheres, respectively. The five most frequently occurring SF3B1 mutation sites are highlighted in bold.

engagement of the intron by U2 snRNP[29]. However, since the location and the interacting-partner of DHX15 in U2 snRNP are yet to be identified, the mechanism of DHX15 in early splicing complex remains an open question.

## Methods

### Transfection, cell lysis and Flag-SF3B1 immunoprecipitation

HEK293F cells were cultured in SMM 293T-II medium (Sino Biological Inc.) under 5% $CO_2$ in a Multitron-Pro shaker (Infors, 130 rpm) at 37 °C. When cell density reached $2.5 \times 10^6$ cells per mL, 1.5 mg of plasmid DNA was transfected into cells using 40-kD linear polyethylenimines (PEIs) (Yeasen). Cells were cultured for another 48 hours for protein expression, harvested in the lysis buffer (20 mM HEPES-KOH, pH 7.9, 150 mM NaCl, 1.5 mM $MgCl_2$ and 0.05% NP-40) supplemented with protease inhibitor cocktail, and disrupted by mild sonication. For SF3B1 immunoprecipitation, the human cDNA of full-length SF3B1 was fused with an N-terminal Flag-tag and cloned into the mammalian expression vector pCAG[51]. After protein expression, the cells were disrupted and the resulting cell lysates were clarified by centrifugation at 20,000 g at 4 °C for 30 min. The cleared lysates were incubated with anti-Flag M2 affinity gel (Sigma) with rotation for 2 hours at 4 °C. The beads were washed five times with lysis buffer and the bound proteins were eluted using 0.4 mg/mL FLAG peptide. Immunoprecipitated proteins were digested overnight with trypsin. The resulting digests were analyzed by mass spectrometry as described[52].

### Preparation of the DDX42-SF3b core complex

Human cDNAs for truncated SF3B1 (residues 454–1304) and full-length SF3B3, PHF5A, SF3B5 and DDX42 were individually cloned into the mammalian expression vector pCAG[51]. SF3B1 (residues 454–1304), SF3B5 and DDX42 were each fused to an N-terminal Flag tag. SF3B3 was fused to an N-terminal Twin-Strep tag. The sequences of all constructs were verified by DNA sequencing. Equal amounts of the plasmids for five components of the DDX42-SF3b complex were co-transfected into HEK293F cells. The cleared cell lysates were loaded to columns containing anti-Flag M2 affinity gel (Sigma) at 4 °C and was allowed to flow through by gravity. The eluent from the anti-Flag columns was applied to Strep-Tactin Sepharose (IBA) for further purification. The target protein complexes were eluted using the lysis buffer supplemented with 50 mM biotin and immediately subjected to size exclusion chromatography (Superdex-200 Increase, GE Healthcare) in SD buffer (20 mM HEPES-KOH, pH 7.9, 150 mM NaCl, 1.5 mM $MgCl_2$). Peak fractions containing the DDX42-SF3b complex were pooled, crosslinked using BS3. Crosslinking was performed by incubating the protein sample with 0.5 mM BS3 (Thermo Fisher) on ice for 1 hour. Then reaction was quenched by addition of 50 mM Tris-HCl, pH 8.0. The crosslinked sample was concentrated to 1.5 mg/mL for cryo-EM analysis.

### Preparation of the human DDX42-U2 complex

The full-length DDX42 was cloned into the pCAG vector with an N-terminal Flag tag. The expression of Flag-DDX42 in HEK293F cells was similar to that of the DDX42-SF3b complex. Briefly, 1.5 mg of the plasmid was pre-incubated with 4 mg of PEIs in 50 ml of fresh medium at room temperature for 20 minutes. The resulting DNA-PEI mixture was then added to one liter of cell culture. Usually, at least two liters of cells were used for each preparation. Cells were collected using the same lysis buffer as that used for the preparation of the DDX42-SF3b

complex and disrupted by mild sonication. After removing cell debris and chromatin aggregates by centrifugation, the supernatant was applied to anti-Flag columns. After extensive washing using the SD buffer (20 mM HEPES-KOH, pH 7.9, 150 mM NaCl, 1.5 mM MgCl₂), proteins were eluted using the FLAG peptide. To remove excessive DDX42 and other contaminants, we further purified the sample by glycerol gradient centrifugation in the presence of 0–0.1% EM-grade glutaraldehyde (Sigma)[27,53]. After centrifugation at 4 °C for 20 hours at 30,000 rpm in a SW32 rotor (Beckman Coulter), the sample was manually collected into 20 fractions from top to bottom. Total RNAs extracted from a 200-μl aliquot for each fraction were analyzed on an 8% denaturing polyacrylamide gel. Based on results of the RNA gel, fractions containing U2 snRNA only were pooled, concentrated using a 100-KDa cut-off Centricon (Millipore), and dialyzed against the SD buffer using a 10-KDa Mini-lyzer (Pierce) for 5 hours. The final sample was concentrated to 0.8 mg/mL for cryo-EM and mass spectrometry analysis.

### Preparation of the DHX15-U2 complex
The full-length DHX15 was cloned into the pCAG vector with an N-terminal Flag tag. Expression of Flag-DHX15 was performed in adherent HeLa cells cultured in SMM 293T-II medium supplemented with 5% FBS. For purification of DHX15-U2 complex, transfected cells from 100 of 15-cm dishes were collected and rinsed once with ice-cold PBS. The following purification procedures are identical to that for preparation of the DDX42-U2 complex. After glycerol gradient centrifugation, the RNA and protein components from each fraction were analyzed on a Urea-PAGE gel and an SDS-PAGE gel, respectively.

### EM specimen preparation and data acquisition
3-μl of the sample was applied to a glow-discharged grid (Quantifoil Au 300 mesh, R1.2/1.3), blotted for 2 seconds and rapidly plunged into liquid ethane using Vitrobot Mark IV (Thermo Fisher Scientific) operating at 8 °C and 100% humidity. Cryo-EM samples were imaged on a 300-kV Titan Krios electron microscope using a nominal magnification of 81,000×. Movies were recorded using a Gatan K3 detector equipped with a GIF Quantum energy filter (slit width 20 eV) at the counted mode, with a pixel size of 0.5435 Å. Data collection were performed using EPU software (Thermo Fisher Scientific). Each stack of 32 frames was exposed for 2.56 seconds, with a dose rate of ~23 counts/second/pixel for each frame. All 32 frames in each stack were aligned and summed using MotionCor2[54] and binned to a pixel size of 1.087 Å. The defocus value for each image varied from −1.5 to −2.0 μm and was determined by Gctf[55].

### Cryo-EM data processing
For the DDX42-SF3b complex, a total of 3,358 micrographs were collected, of which 3,051 micrographs were selected for further processing. A total of 1,282,572 particles were picked using Gautomatch (developed by Kai Zhang, https://www2.mrc-lmb.cam.ac.uk/research/locally-developed-software/zhang-software/#gauto) and subjected to three-dimensional (3D) classification. The initial 3D volume of the DDX42-SF3b complex was generated from the previously reported cryo-EM map of the SF3b complex[31]. To avoid losing good particles, we simultaneously performed two parallel runs of single-reference 3D classification (Round 1). After Round 1, good particles were selected and merged, and duplicated particles were removed. The remaining 786,305 particles were applied to another two parallel runs of single-reference 3D classification (Round 2), but with 2× binned particles (pixel size: 2.174 Å). Using re-centered and re-extracted particles (pixel size: 1.087 Å), a reconstruction of the DDX42-SF3b complex at an average resolution of 3.3 Å was generated. Then, an additional round (Round 3) of 3D classification was performed. The remaining particles were classified with a soft mask on the core region of the DDX42-SF3b complex. The good class containing 234,800 particles (83.2% of the

input) yielded a reconstruction at an average resolution of 2.6 Å. These particles were further locally classified and refined using different soft masks, generating two additional reconstructions for the DDX42 N-terminus and DDX42 helicase domain at resolutions of 3.3 Å and 3.2 Å, respectively.

For DDX42-U2 complex, a total of 5,439 micrographs were collected, of which 5,229 micrographs were selected for further processing, from which 2,698,248 particles were generated using Gautomatch. After particle extraction, three rounds of two-dimensional (2D) classification were performed using cryoSPARC[56], which gave rise to a final data set of 1,036,422 good particles. This data set was then used for generation of an initial model and further data processing. Through several rounds of hetero-refinement and non-uniform refinement, a final reconstruction at 2.7 Å for the DDX42-U2 complex was obtained from 430,443 particles. Similarly, 99,918 particles from 2,325 micrographs yielded a reconstruction of approximate 5 Å for the for DHX15-U2 complex.

Reported resolution limits were calculated on the basis of the FSC 0.143 criterion with a high-resolution noise substitution method[57]. Prior to visualization, all EM maps were corrected for modulation transfer function (MTF) of the detector, and then sharpened by applying a negative B-factor that was estimated using automated procedure[58]. Local resolution variations were estimated using ResMap[59].

### Model building and refinement
We combined de novo model building and rigid docking of components with known structures to generate the atomic models. Identification and docking of the components of the DDX42-SF3b complex and U2 snRNP were facilitated by the structures of human SF3b complex[31] and U2 snRNP[36]. Protein components derived from known structures are summarized in Supplementary Table 2. These structures were fitted into density using CHIMERA[60] and manually adjusted according to the density map using Coot[61]. Briefly, the atomic coordinates of the SF3b core complex (PDB code: 5ZYA) were directly docked into our 2.6-Å density map of the DDX42-SF3b complex. After this step, there are three EM density patches in the core region and a dumbbell-shaped EM density lobe at the peripheral region remained unassigned. Guided by secondary structure prediction, de novo model building was performed for these unassigned EM densities. The N-terminal sequences of DDX42 fit well with the EM density patches at the core region. The helicase domain of DDX42 was docked into the dumbbell-shaped EM density lobe at the periphery and manually adjusted. For DDX42-U2, the coordinates from 17S U2 snRNP (PDB code: 7EVO)[23] were fitted into our 2.6-Å EM map and manually adjusted for each component. The N-terminus of DDX42 and the RRM and Linker domain of TAT-SF1 were de novo modeled on the basis of our high-resolution map.

The final models of the DDX42-SF3b complex and DDX42-U2 complex were refined according to the high quality cryo-EM maps using REFMAC in reciprocal space[62] and secondary structure restraints that were generated by ProSMART[63]. Overfitting of the model was monitored by refining the model in one of the two independent maps from the gold-standard refinement approach, and testing the refined model against the other[64]. The structures of the DDX42-SF3b complex and DDX42-U2 were validated through examination of the Molprobity scores and statistics of the Ramachandran plots. Molprobity scores were calculated as described[65].

### Preparation of Flag-DDX42 and Flag-DDX46
The Flag-DDX42 and Flag-DDX46 were overexpressed in HEK293F cells. Cells were harvested with high-salt lysis buffer (20 mM HEPES-KOH, pH 7.9, 500 mM NaCl, 1.5 mM MgCl₂ and 0.05% NP-40) and disrupted by sonication. Target proteins were selected by anti-Flag M2 affinity gel (Sigma) in SD buffer (20 mM HEPES-KOH, pH 7.9, 150 mM

NaCl, 1.5 mM $MgCl_2$) and further purified through ion-exchange chromatography using a Hitrap Heparin column (GE Healthcare). The peak fractions were pooled, concentrated, and used for binding assays.

## Measurement of binding affinity

We employed bio-layer interferometry to measure the binding kinetics between the SF3b core and the RNA helicase DDX42 or DDX46. The SF3b core was prepared as described above, except that the Strep-tag on SF3B3 was replaced by a hexa-histidine (6xHis) tag. The purified SF3b core (6xHis-tagged) was diluted to 20 µg/mL in the lysis buffer and captured on Ni-NTA biosensors until the thickness signal reaches 1.0 nm. After loading, the sensors were washed for 180 seconds in the lysis buffer and then immersed into wells containing Flag-DDX42 or Flag-DDX46 proteins for about 100 seconds (association phase). Then, the sensors were immersed into new wells containing the lysis buffers for an additional 100 seconds (dissociation phase). The background signal was recorded using a reference sensor with 6xHis-SF3b loaded, but no analyte protein during the association phase. Curve fitting was performed using a 1:1 binding model with the ForteBio data analysis software. The mean on-rate ($K_{on}$), off-rate ($K_{off}$), and dissociation constant ($K_D$) values were determined by averaging all binding curves that matched the theoretical fit with an $R^2$ value of 0.95.

## Competition assay between DDX42 and DDX46

The SF3b core was prepared similarly as for the DDX42-SF3b core complex, except that the plasmid for Flag-DDX42 was omitted during cell transfection. For competition assays, the purified Flag-DDX42 and SF3b core complex were incubated at a molar ratio of 1:1.2 in the lysis buffer for 30 minutes to allow complex formation. Then, two or ten-fold molar excess of purified Flag-DDX46 was added to the above solution and incubated for another 30 minutes. The resulting protein mixtures were applied to Strep-Tactin affinity selection, and the selected proteins were analyzed on SDS-PAGE. Conversely, in the case of using DDX42 to compete DDX46 from the SF3b complex, purified Flag-DDX46 and SF3b core complex (1:1.2) were incubated for 30 minutes, followed by addition of excess Flag-DDX42. The mixture was incubated for 30 minutes before loading onto Strep-Tactin column. The eluent was analyzed by SDS-PAGE. Proteins were verified by western blot and mass spectrometry.

## Reporting summary

Further information on research design is available in the Nature Portfolio Reporting Summary linked to this article.

# Data availability

The atomic coordinates for DDX42-SF3b complex and the DDX42-U2 complex have been deposited in the Protein Data Bank (PDB) under the accession code PDB-7EVN and PDB-8HK1, respectively. The cryo-EM maps of the core region of DDX42-SF3b complex, DDX42 N-plug, DDX42 helices α$_{-3}$/α$_{-2}$/α$_{-1}$ and DDX42 helicase domain have been deposited in the Electron Microscopy Data Bank (EMDB) with the accession codes EMD-31330, EMD-31331, EMD-31332 and EMD-31333, respectively. The cryo-EM maps for DDX42-U2 and DHX15-U2 complex have been deposited in the EMDB with the accession codes EMD-34841 and EMD-34845, respectively. All other data are available from the corresponding author upon request. Source data are provided with this paper.

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

## Acknowledgements

We thank the Cryo-EM Facility, the Computing Center and the Crystallography Facility of Westlake University for facility support. We thank Shaoming Feng and Jinheng Pan at the Mass Spectrometry & Metabolomics Core Facility of Westlake University for technical assistance. This work was supported by funds from the National Natural Science Foundation of China (31930059 to Y.S.), the China Postdoctoral Science Foundation (2020M671806 to Xia.Z), the National Postdoctoral Program

for Innovative Talents of China (BX20200305 to Xia.Z), and Start-up funds from Westlake University (to Y.S.).

## Author contributions

F.Y., Z.C. and Xia.Z designed the strategies to purify the DDX42-SF3b complex, the human DDX42-U2 complex and the DHX15-U2 complex. F.Y. performed most of the experiments. Xia.Z, F.Y., T.B. and Z.X. prepared cryo-EM samples and collected the EM data. Xia.Z, Xie.Z and N.A.L. processed the EM data, calculated the EM map, and built the atomic models. Y.S. supervised the project. All authors contributed to data analysis. Y.S. and Xia.Z wrote the manuscript.

## Competing interests

N.A.L. is an employee of Foghorn Therapeutics, Inc. The remaining authors declare no competing interests.
