## [Peer Review File · Nature Communications]

Mechanisms of the RNA Helicases DDX42 and DDX46 in Human U2 snRNP AssemblyREVIEWER COMMENTS

Reviewer #1 (Remarks to the Author):

Yang et al. have determined cryo-EM structures of three U2-related complexes that contain the RNA helicase DDX42 or DHX15. The DDX42-SF3b complex was assembled by co-expressing DDX42 with the U2 components, SF3B1, SF3B3, SF3B5 and PHF5A, and it was purified through tandem affinity chromatography and gel filtration. A putative precursor of the 17S U2 snRNP was isolated via Flag-tagging of DDX42, and a DHX15-U2 complex was isolated via Flag-tagging of DHX15, from HEK293 cells. Although the cryo-EM map of the DHX15-U2 complex failed to reveal critical interactions between DHX15 and the U2 snRNP, the DDX42-SF3b and pre-17S U2 snRNP structures provide detailed information on the interaction between DDX42 and the SF3b complex.

DDX42 has been shown to co-precipitate with SF3B1 from cell extracts, but its function was largely unknown. The cryo-EM structures of DDX42-SF3b and the pre-17S U2 snRNP presented in this study reveal how DDX42 is anchored on SF3B1 via distinct N-terminal motifs, including an N-plug and three α -helices. The N-plug is placed in the RNA path of a positively-charged groove, i. e., the site on the 17S U2 snRNP occupied by the acidic loop of DDX46 and where the polypyrimidine tract of pre-mRNA binds on the spliceosome. The authors have also demonstrated that DDX42 and DDX46 bind competitively to SF3b. Based on these results, they propose a model of sequential engagement of the RNA path by DDX42, DDX46 and the pre-mRNA during formation of the 17S U2 snRNP and of the prespliceosome.

The DDX42-SF3b and pre-17S U2 snRNP structures described in this work undoubtedly provide valuable information on the possible function of DDX42, and how DDX42 and DDX46 might coordinate to mediate formation of the 17S U2 snRNP. The results largely support the model they propose, but several issues need to be clarified. There are also quite a few errors in writing and citations.

1. Fig. 1a shows that DDX42 and DDX46 coprecipitated stoichiometrically with SF3B1, but DHX15 is less abundant in the precipitate. The authors suggest that DHX15 binds to SF3B1 less strongly. However, it is also possible that the DHX15-associated SF3B1 complex is less abundant, given that SF3B1 associates with various complexes, rather than reflecting any difference in binding affinity.
2. It should be explained why the DDX42-SF3b complex was reconstituted by co-expressing DDX42 with only four, and not all components, of the SF3b complex, and if doing so truly represents the SF3b complex.
3. P.4, line 72: Recent studies have revealed "the" both.....
4. p.5, line 102: The description here already assumes that the DDX42-associated complex is an assembly precursor of the 17S U2 snRNP before providing experimental evidence.
5. p.8, line 173: "known" as branchpoint-interacting stem loop
6. Extended Data Fig. 3a: Proteins of the purified pre-17S complex should be marked to see which proteins are identifiable in the purified fraction. Marker sizes should also be indicated.
7. In Fig. 3a, it is not clear to me how the DDX42 helicase domain is assigned to the position circled by the dashed line. The corresponding location is occupied by UHM of TAT-SF1 in the 17S U2 snRNP, in which the DDX46 helicase domain lies on top of TAT-SF1.
8. P. 9, line 183: Many RNA helicases have both ATP-dependent and ATP-independent functions. It is not known if the ATPase or helicase activity of DDX42 is required for U2 snRNP assembly. It is too speculative to say that the pre-17 U2 snRNP complex represents a state after DDX2 has fulfilled its RNA helicase activity but is yet to be released.
9. P.10, line 196: contribute to "maintaining" the BSL conformation.
10. P12, line 241: the wording here is difficult to decipher. Please revise.
11. P.12, line 246: "stabilizes".
12. P.12, line 251: There were no crosslinking experiments conducted in ref. 41.
13. P.13, line 275: "Apart" from DDX42.....

14. P.13, line 277: Extended Data Fig. 4a does not support the description.
15. P.13, line 282: Citing Fig. 5a here is meaningless.
16. In Figure 5a, it should be explained why TAT-SF1 is drawn differently as a free form structure.

Reviewer #2 (Remarks to the Author):

Yang et al. present high-resolution cryoEM reconstructions of DDX42-SF3b and pre-17S U2 snRNP, revealing how DDX42 docks to the particles via three motifs in its disordered N-terminal region. Structural comparisons and (competitive) binding studies reveal that binding of DDX42 to SF3B1 is mutually exclusive with DDX46 and, most likely, with the PPT of pre-mRNA. The authors also present a lower resolution cryoEM reconstruction of DHX15-modified U2 snRNP, in which they did not observe any major DHX15-dependent structural changes and in which DHX15 could not be localized, as it may be bound flexibly to peripheral regions. The authors also map the positions of cancer-linked residue substitutions to the structures, supporting the notion that such residue substitutions reduce DDX42 and DDX46 binding, perhaps leading to premature opening of an RNA path on SF3b. The results thus suggest that, apart from guiding U2 snRNA rearrangements during U2 snRNP assembly, the helicases safeguard against aberrant/premature interactions of SF3b with (non-cognate) RNAs or other proteins.

The results presented are novel and interesting, expand our knowledge of the molecular mechanisms underlying U2 snRNP assembly and provide insights into possible molecular disease principles. The work appears to be technically sound and the manuscript is in general clearly written.

Specific comments:

1. In Figure 5b, cancer-related residue substitution that affect DDX42 only, DDX46 only or both should be indicated, e.g. by different colors.
2. While it is obvious how substitution of Asp894, Tyr898 and Glu902 can selectively affect DDX46 binding, it is not clear from the structural analysis or the current presentation how certain cancer-related residue substitutions can selectively affect DDX42 (it seems as if DDX46 binding should also be affected in each case?). Could the authors explain and illustrate the effects of these residue substitutions in more detail?
3. Related to the above, it would be nice if the authors could select a couple of DDX42-specific and DDX46-specific cancer-related residue exchanges and, using their established binding assay, show that the SF3b containing the respective SF3B1 variants indeed selectively affect DDX42 or DDX46 binding in vitro.
4. Structural elements of the DDX42 N-terminal anchoring region should be labeled in Figure 1b-d.
5. Methods should contain more details of the crosslinking strategies used to stabilize both DDX42-SF3b and pre-17S U2 snRNP for cryoEM studies, including specific buffers used and crosslinker concentrations.
6. Methods used to prepare Flag-tagged DDX42 and DDX46 for interaction studies should be described.
7. Out of curiosity – DDX42-SF3b and pre-17S U2 snRNP were produced in / obtained from HEK293F cells, while adherent HeLa cells were used for preparation of DHX15-U2 snRNP. Why the different cell lines for the different complexes?

Reviewer #3 (Remarks to the Author):

The manuscript "Mechanisms of the RNA Helicases DDX42 and DDX46 in Human U2 snRNP Assembly" by Yang et al, reports the interactions and structures of RNA-dependent ATPases with components of the human spliceosome. One major conclusion is that DDX42 interacts with the SF3b complex in a manner similar to DDX46, which is supported by data showing mutually exclusive binding. Another conclusion is that SF3B1 cancer mutations map to one of the interaction sites that also contact pre-mRNA during spliceosome assembly. The authors speculate that different effects on splicing by the mutations may be independently mediated during either U2 snRNP assembly defects (DDX42) or branch point sequence recognition (DDX46). Overall, the paper is communicated clearly with informative figures that highlight important features of the structural models very well. However, before publication, the authors need to address a few major and minor issues.

Major issues:

1. For the DDX42-SF3b complex, FLAG-tags present on SF3B1, SF3B5 and DDX42 were used to isolate complexes from HEK293F cells. How did the authors ensure that endogenous U2 snRNP-associated proteins were not present in the complex? Considering the lower resolution of the DDX42 portion of the map and poor fit of the DDX42 model indicated by the PDB validation report, I think that the possibility of SF3B-complexes that also co-purifies with DDX46 or other proteins cannot be ruled out. The density used to model DDX46 N-terminus and pre-mRNA in previous structures also does not appear to be strong, which raises some ambiguity about the proposed interactions that are being compared. This issue could be addressed in part by showing the tagged protein levels relative to wildtype or providing mass spectrometry data.
2. I have similar concern with the DDX42-17S U2 snRNP purification. Based on ExFig3A, DDX42 eluent contains multiple RNA molecules (what stain is being used for this gel?). No data is shown to ensure that DDX42 is the only helicase associated with U2 snRNA in gradient fractions or that U2 snRNP association with FLAG-tagged DDX42 is direct. An incompletely annotated PDB validation report (the identity of individual chains is not designated) makes it very challenging to comment on the quality of the model and fit to the EM density. At the least, the density used to model DDX42 residues should be included similar to ExFig5.
3. What evidence justifies the claim that the DDX42-17S U2 snRNP is an assembly intermediate and precedes bona fide 17S U2 snRNP assembly? Could the interaction with U2 snRNP result from DDX42 overexpression? Could the complex be a disassembly intermediate?

Minor issues:

1. Can the authors explain how the data shown in Figure 1A indicates that both DDX42 and DDX46 can be stoichiometrically associated with SF3B1, especially when their binding is presumably mutually exclusive? The experimental method for this experiment is missing and should be included. The authors also use this data to conclude that DHX15 has weaker binding to U2 snRNP. Can they rule out the alternative explanation that DHX15 association with U2 snRNP is regulated, and only a subset of the U2 snRNP present in cells would be found in the conformation to which DHX15 binds?
- 2 The authors state that the so-called "loose" and "weak" interaction of the DDX42 helicase domain with the SF3b core is required by its activity as an RNA helicase for translocation. The authors should be careful when relating weak density to binding affinity, because the helicase domain may instead be interacting with one of the more flexible/disordered regions of the SF3b. complex. Furthermore reference 20, which is cited to point out a so-called "RNA chaperone" activity for DDX42, provides evidence that DDX42 is an RNA-dependent ATPase that does not translocate to unwind RNA, but instead uses ATP-hydrolysis to regulate binding and release of the protein from single stranded RNA. The protein can also disrupt RNA/protein interactions Emphasizing that the target of DDX42 activity in

the context of U2 snRNP assembly or function is unknown would be helpful for readers.

3. The methods state that the cell line utilized is Hek293F, but results refer to them as Hek293. This should be rectified to match as they are different lines.

4. For figure 1D, labels of the different protein components would be helpful as the yellow from the EM map distorts the underlying colors. Alternatively, a gray map would

5. For Figure 3, "TATSF1" should be included in the labels for its different domains, which should have the same color.

6. The manuscript has a number grammatical and spelling errors. A few of these are highlighted below:

Line 44- "spliceosome" should be preceded by an article. Typically "the spliceosome" is used.

Line 119- DXH15 should be DHX15

Line 173- knowing to known

Point-by-Point response to reviewers' comments

We thank the reviewers for the constructive comments they provided. We have answered all questions in detail and revised the manuscript according to the suggestions.

Reviewer #1:

Yang et al. have determined cryo-EM structures of three U2-related complexes that contain the RNA helicase DDX42 or DXH15. The DDX42-SF3b complex was assembled by co-expressing DDX42 with the U2 components, SF3B1, SF3B3, SF3B5 and PHF5A, and it was purified through tandem affinity chromatography and gel filtration. A putative precursor of the 17S U2 snRNP was isolated via Flag-tagging of DDX42, and a DXH15-U2 complex was isolated via Flag-tagging of DXH15, from HEK293 cells. Although the cryo-EM map of the DXH15-U2 complex failed to reveal critical interactions between DXH15 and the U2 snRNP, the DDX42-SF3b and pre-17S U2 snRNP structures provide detailed information on the interaction between DDX42 and the SF3b complex.

DDX42 has been shown to co-precipitate with SF3B1 from cell extracts, but its function was largely unknown. The cryo-EM structures of DDX42-SF3b and the pre-17S U2 snRNP presented in this study reveal how DDX42 is anchored on SF3B1 via distinct N-terminal motifs, including an N-plug and three α -helices. The N-plug is placed in the RNA path of a positively charged groove, i.e., the site on the 17S U2 snRNP occupied by the acidic loop of DDX46 and where the polypyrimidine tract of pre-mRNA binds on the spliceosome. The authors have also demonstrated that DDX42 and DDX46 bind competitively to SF3b. Based on these results, they propose a model of sequential engagement of the RNA path by DDX42, DDX46 and the pre-mRNA during formation of the 17S U2 snRNP and of the prespliceosome.

The DDX42-SF3b and pre-17S U2 snRNP structures described in this work undoubtedly provide valuable information on the possible function of DDX42, and how DDX42 and DDX46 might coordinate to mediate formation of the 17S U2 snRNP. The results largely support the model they propose, but several issues need to be clarified. There are also quite a few errors in writing and citations.

Response: We thank reviewer #1 for recognizing the significance of our work. We have carefully reviewed and responded to his/her concerns point-by-point in the sections below.

Fig. 1a shows that DDX42 and DDX46 coprecipitated stoichiometrically with SF3B1, but DXH15 is less abundant in the precipitate. The authors suggest that DXH15 binds to SF3B1 less strongly. However, it is also possible that the DXH15-associated SF3B1 complex is less abundant, given that SF3B1

associates with various complexes, rather than reflecting any difference in binding affinity.

Response: Point accepted. It's possible that DHX15-associated SF3B1 complex is intrinsically less abundant than that of DDX42 and DDX46. We removed this inappropriate statement in the revised manuscript.

It should be explained why the DDX42-SF3b complex was reconstituted by co-expressing DDX42 with only four, and not all components, of the SF3b complex, and if doing so truly represents the SF3b complex.

Response: We thank the reviewer for this comment. In fact, we have put a lot of efforts to co-express all components of SF3b complex, however, not all of these proteins could be recombinantly expressed. To our knowledge, the whole SF3b complex has not been successfully reconstituted so far. The combination of four components used here is regarded as the “core” of SF3b and has been chosen by several groups in the splicing field, such as Luhrmann group (*Cretu et al., Molecular Cell, 2016*), Pena group (*Cretu et al., Molecular Cell, 2018*) and our group (*Finci and Zhang et al., Genes & Development, 2018*). Furthermore, the SF3b core adopts nearly identical conformation in the structures of DDX42-SF3b complex and 17S U2 snRNP. In this regard, we believe our structure could faithfully represent the interactions between DDX42 and SF3b complex.

P.4, line 72: Recent studies have revealed “the” both....

Response: We apologize for this typo. During the revision, we carefully checked our manuscript and corrected these errors.

p.5, line 102: The description here already assumes that the DDX42-associated complex is an assembly precursor of the 17S U2 snRNP before providing experimental evidence.

Response: Point accepted. To avoid misunderstandings, we modified the description in this section and explained why we thought the DDX42-associated complex is an assembly precursor of 17S U2 by adding this sentence: “*Because DDX46 is yet to be recruited, the purified DDX42-associated complex likely represents an assembly precursor of 17S U2 snRNP (thus referred to as pre-17S U2).*”

p.8, line 173: “known” as branchpoint-interacting stem loop...

Response: This was corrected in the revised version.

Extended Data Fig. 3a: Proteins of the purified pre-17S complex should be marked to see which proteins are identifiable in the purified fraction. Marker sizes should also be indicated.

Response: We thank the reviewer for this suggestion. We agree that an SDS-PAGE gel showing the protein components of pre-17S U2 complex

would be very informative. However, due to the presence of 0-0.1% glutaraldehyde during glycerol gradient centrifugation, the protein samples were chemically crosslinked and thus stacked in the wells during gel electrophoresis. Instead, we performed mass spectrometric analyses on the purified pre-17S U2 complex (provided as Extended Data Table 3). Proteins of pre-17S U2 snRNP identified by our cryo-EM map were indicated.

In Fig. 3a, it is not clear to me how the DDX42 helicase domain is assigned to the position circled by the dashed line. The corresponding location is occupied by UHM of TAT-SF1 in the 17S U2 snRNP, in which the DDX46 helicase domain lies on top of TAT-SF1.

Response: We apologize for this confusion. Compared to 17S U2 snRNP, in which the UHM domain of TAT-SF1 is clearly identified, there's no obvious EM density for the UHM domain in pre-17S U2 complex (Extended Data Fig. 7b). This suggests the UHM domain is relative mobile in pre-17S U2 complex and may be stabilized through interactions with DDX46 upon formation of 17S U2 snRNP. Although the exact location of DDX42 helicase domain is unclear in pre-17S U2, we identified an EM density lobe that lies on top of the BSL of U2 snRNA (see Extended Data Fig. 7c). It's possible that part of this lobe is contributed by the UHM domain, but given the much larger size, there should be additional protein elements. We are not sure whether this lobe belongs to DDX42 helicase domain, because the poor local resolution does not allow reliable docking of known structures. However, we believe this is an important information, so we indicated this possible location of DDX42 helicase domain using dashed lines.

P. 9, line 183: Many RNA helicases have both ATP-dependent and ATP-independent functions. It is not known if the ATPase or helicase activity of DDX42 is required for U2 snRNP assembly. It is too speculative to say that the pre-17 U2 snRNP complex represents a state after DDX2 has fulfilled its RNA helicase activity but is yet to be released.

Response: Point accepted. As the reviewer pointed out, although our study suggests DDX42 may play essential roles during U2 snRNP assembly, it is unclear if the ATPase or helicase activity is indeed required for this process. To avoid controversies, we rephrased this sentence as: *"Our pre-17S U2 likely represents an intermediate state in which DDX42 has fulfilled its function but is yet to be released"*.

P.10, line 196: contribute to "maintaining" the BSL conformation.

Response: This is corrected in the revised version.

P12, line 241: the wording here is difficult to decipher. Please revise.

Response: We apologize for the confusion. We rephrased this section in the revised manuscript.

P.12, line 246: “stabilizes”.

Response: This is corrected in the revised version.

P.12, line 251: There were no crosslinking experiments conducted in ref. 41.

Response: We apologize for this error of citation. The ref. 41 here should be changed to ref. 36 (*Zhang et al., Nature, 2020*). In this paper, crosslinks are observed between the UHM domain of TAT-SF1 with DDX46 (see Extended Data Fig. 7c from ref. 36).

P.13, line 275: “Apart” from DDX42...

Response: This is corrected in the revised version.

P.13, line 277: Extended Data Fig. 4a does not support the description.

Response: We thank the reviewer for this observation. Extended Data Fig. 4b instead of 4a should be cited here. In the protein sample of purified DHX15-U2 complex, several G-patch proteins, exemplified by TFIP11 and CHERP, were identified by mass spectrometric analyses.

P.13, line 282: Citing Fig. 5a here is meaningless.

Response: Point accepted. We removed this citation here.

In Figure 5a, it should be explained why TAT-SF1 is drawn differently as a free form structure.

Response: We appreciate this query. TAT-SF1 contains three domains: the RRM, Linker, and UHM domain. The Linker domain and UHM domain shield the tip of BSL in the structure of 17S U2 snRNP (Fig. 4a middle right). But in pre-17S U2, the exact position of UHM domain is unclear as there's no obvious EM density on the right side of BSL (Extended Data Fig. 7b). We suppose that UHM domain is relative mobile in pre-17S U2 and may be stabilized through interactions with DDX46 in 17S U2 snRNP. Therefore TAT-SF1 is drawn differently in pre-17S U2 as to the free form structure. To avoid misunderstandings, we added a dashed circle to indicate the potential position of the UHM domain in the cartoon model of pre-17S U2 and explained this drawing in the figure legends.

Reviewer #2:

Yang et al. present high-resolution cryo-EM reconstructions of DDX42-SF3b and pre-17S U2 snRNP, revealing how DDX42 docks to the particles via three motifs in its disordered N-terminal region. Structural comparisons and competitive binding studies reveal that binding of DDX42 to SF3B1 is mutually exclusive with DDX46 and, most likely, with the PPT of pre-mRNA. The authors also present a lower resolution cryo-EM reconstruction of DHX15-modified U2 snRNP, in which they did not observe any major DHX15-dependent structural changes and in which DHX15 could not be localized, as it may be bound flexibly to peripheral regions. The authors also map the positions of cancer-linked residue substitutions to the structures, supporting the notion that such residue substitutions reduce DDX42 and DDX46 binding, perhaps leading to premature opening of an RNA path on SF3b. The results thus suggest that, apart from guiding U2 snRNA rearrangements during U2 snRNP assembly, the helicases safeguard against aberrant/premature interactions of SF3b with non-cognate RNAs or other proteins.

The results presented are novel and interesting, expand our knowledge of the molecular mechanisms underlying U2 snRNP assembly and provide insights into possible molecular disease principles. The work appears to be technically sound and the manuscript is in general clearly written.

Response: Reviewer #2 commented favorably on our manuscript and only raised several specific points for us to address.

In Figure 5b, cancer-related residue substitution that affect DDX42 only, DDX46 only or both should be indicated, e.g. by different colors.

Response: We appreciate this suggestion. The figure has been modified as suggested.

While it is obvious how substitution of Asp894, Tyr898 and Glu902 can selectively affect DDX46 binding, it is not clear from the structural analysis or the current presentation how certain cancer-related residue substitutions can selectively affect DDX42 (it seems as if DDX46 binding should also be affected in each case?). Could the authors explain and illustrate the effects of these residue substitutions in more detail?

Response: We apologize for this confusion. As listed in Extended Data Table 4, the SF3B1 residues that interact with DDX42 or DDX46 are grouped into three classes: with DDX42 only, with DDX46 only, and with both. For those residues that interact with both DDX42 and DDX46 (such as K700, H662), we do not think their substitutions have selectivity on DDX42 or DDX46. In fact, these cancer-related mutations, which clustered in the RNA path, were shown to abrogate the binding of both DDX42 (*ref. 46: Zhao et al., J Biochem., 2022*) and DDX46 (*ref. 23: Zhang et al., bioRxiv, 2022*). The selectivity on DDX42 and DDX46 by cancer-related mutations is not mentioned in our manuscript,

thus we believe this is a misunderstanding. During revision, we clarified this point by adding a sentence in this section: “*Substitutions of these residues in cancers abrogate the binding of both DDX42 and DDX46*”.

Related to the above, it would be nice if the authors could select a couple of DDX42-specific and DDX46-specific cancer-related residue exchanges and, using their established binding assay, show that the SF3b containing the respective SF3B1 variants indeed selectively affect DDX42 or DDX46 binding *in vitro*.

Response: We thank the reviewer for this suggestion. As mentioned above, the effects of SF3B1 cancer mutations on DDX42 and DDX46 have been extensively discussed by Zhao et al. (ref. 46) and a previous study from our group (ref. 23), in which both binding assays and RNA-sequencing analyses were provided. In this manuscript, we focused more on the competitive SF3B1 binding by DDX42 and DDX46; we quantitatively measured the relative binding affinities using bio-layer interferometry. In addition, we also performed *in vitro* competitive assays for DDX42 and DDX46. We believe our structural and biochemical observations should shed light on the mechanisms of DDX42 and DDX46 during U2 snRNP assembly.

Structural elements of the DDX42 N-terminal anchoring region should be labeled in Figure 1b-d.

Response: Point accepted. We modified the figure as suggested.

Methods should contain more details of the crosslinking strategies used to stabilize both DDX42-SF3b and pre-17S U2 snRNP for cryo-EM studies, including specific buffers used and crosslinker concentrations.

Response: Point accepted. We provided these details in the revised manuscript.

Methods used to prepare Flag-tagged DDX42 and DDX46 for interaction studies should be described.

Response: Point accepted. Detailed methods were provided as requested.

Out of curiosity – DDX42-SF3b and pre-17S U2 snRNP were produced in / obtained from HEK293F cells, while adherent HeLa cells were used for preparation of DHX15-U2 snRNP. Why the different cell lines for the different complexes?

Response: We thank the reviewer for this interesting observation. We chose HEK293F cells for protein production because the cell amounts could be easily scaled up by suspension culture, which helps to obtain enough protein. However, when we tried to pulldown DHX15-related complexes in HEK295F cells, a lot of “contaminants” were detected in the purified sample. Most of these “contaminants” are ribosome-related proteins, which is not surprising as

DHX15 is involved in ribosome biogenesis. This sample hinders determination of the structure. Unexpectedly, in an un-related study, we found the abundance of ribosome-related proteins was largely decreased if we perform the DHX15-pulldown experiment in adherent HeLa cells. We have no idea whether this is due to a lower expression level of DHX15 in HeLa cells, thereby reducing non-specific or weak binding, or simply a result of different cell types. We believe this is an interesting observation and worth following up investigation. Anyway, to obtain pure and enough DHX15-U2 complex for mass-spec and cryo-EM study, we utilized HeLa cells for sample preparation. Notably, 100 of 15-cm dishes of HeLa cells were used for just a batch of purification.

Reviewer #3:

The manuscript “Mechanisms of the RNA Helicases DDX42 and DDX46 in Human U2 snRNP Assembly” by Yang et al, reports the interactions and structures of RNA-dependent ATPases with components of the human spliceosome. One major conclusion is that DDX42 interacts with the SF3b complex in a manner similar to DDX46, which is supported by data showing mutually exclusive binding. Another conclusion is that SF3B1 cancer mutations map to one of the interaction sites that also contact pre-mRNA during spliceosome assembly. The authors speculate that different effects on splicing by the mutations may be independently mediated during either U2 snRNP assembly defects (DDX42) or branch point sequence recognition (DDX46). Overall, the paper is communicated clearly with informative figures that highlight important features of the structural models very well. However, before publication, the authors need to address a few major and minor issues.

Response: We thank reviewer #3 for recognizing the significance of our work. We have addressed his/her concerns point-by-point in the sections below.

For the DDX42-SF3b complex, FLAG-tags present on SF3B1, SF3B5 and DDX42 were used to isolate complexes from HEK293F cells. How did the authors ensure that endogenous U2 snRNP-associated proteins were not present in the complex? Considering the lower resolution of the DDX42 portion of the map and poor fit of the DDX42 model indicated by the PDB validation report, I think that the possibility of SF3B-complexes that also co-purifies with DDX46 or other proteins cannot be ruled out. The density used to model DDX46 N-terminus and pre-mRNA in previous structures also does not appear to be strong, which raises some ambiguity about the proposed interactions that are being compared. This issue could be addressed in part by showing the tagged protein levels relative to wildtype or providing mass spectrometry data.

Response: We thank the reviewer for this concern. **First**, the expression levels of recombinant proteins are much higher than endogenous ones. **Second**, to purify the DDX42-SF3b complex, we performed **two-step affinity purification**: first, FLAG-tags on SF3B1, SF3B5 and DDX42 were used for affinity selection; then, the sample were further purified through a Strep-tag on SF3B3. This two-step selection strategy could largely reduce the “contaminants” from endogenous protein. **Third**, to further address this point, protein components of purified DDX42-SF3b complex were analyzed through mass spectrometry (see the table below). As expected, DDX46 and other components of U2 snRNP (except DHX15) are nearly undetectable. Consequently, it’s unlikely that the relative low resolution of DDX42 portion is caused by the presence of DDX46 in the sample. **Most importantly**, the EM map quality for the N-terminus of DDX42 is sufficient for assignment of most side chains; only the helicase domain of DDX42 is less-well resolved

(Extended Data Fig. 5). The proposed interactions between the N-terminus of DDX42 and SF3B1 are reliable.

Description	PEP Score	Coverage [%]	# Peptides
Splicing factor 3B subunit 3	1087.13	68	88
ATP-dependent RNA helicase DDX42	804.953	72	60
Splicing factor 3B subunit 1	726.941	46	58
Splicing factor 3B subunit 5	82.708	65	7
Heat shock 70 kDa protein 1B	81.079	32	14
PHD finger-like domain-containing protein 5A	55.788	70	8
Heat shock cognate 71 kDa protein	52.159	20	11
Tubulin beta chain	38.127	30	10
Tubulin alpha chain	31.263	21	9
WD repeat-containing protein 70	28.362	13	7
Coiled-coil domain-containing protein 97	23.977	21	5
WD repeat-containing protein 61	23.836	20	5
Endoplasmic reticulum chaperone BiP	21.353	13	7
Tetratricopeptide repeat protein 33	16.283	20	4
Kanadaplin	16.145	8	5
Ubiquitin-40S ribosomal protein S27a	14.376	44	4
Splicing factor 1	12.905	5	3
RNA-binding region-containing protein 3	11.667	6	3
Sodium channel modifier 1	9.501	16	3
ATP-dependent RNA helicase DHX15	2.902	2	1

I have similar concern with the DDX42-17S U2 snRNP purification. Based on ExFig3A, DDX42 eluent contains multiple RNA molecules (what stain is being used for this gel?). No data is shown to ensure that DDX42 is the only helicase associated with U2 snRNA in gradient fractions or that U2 snRNP association with FLAG-tagged DDX42 is direct. An incompletely annotated PDB validation report (the identity of individual chains is not designated) makes it very challenging to comment on the quality of the model and fit to the EM density. At the least, the density used to model DDX42 residues should be included similar to ExFig5.

Response: We appreciate the reviewer for this concern. The RNA gel from ExFig3a was stained by SYBR-Gold, a commonly used RNA staining reagent. As the reviewer pointed out, there are multiple bands of RNA from DDX42 eluent. In fact, some of these bands, which we believe are ribosomal RNAs, are frequently seen on the RNA gels of spliceosome sample or other complexes purified from the nuclear extract. It's currently unknown if DDX42 is involved in ribosome biogenesis, but indeed ribosomes are commonly identified contaminants during spliceosomal complex purification. Presence of ribosomes in the DDX42-eluent can also be verified by the mass-spec data (see Extended Data Table 3). In addition, we prepared a figure (Extended Data Fig. 7) to show the features of EM map for the pre-17S U2 snRNP as the reviewer requested.

What evidence justifies the claim that the DDX42-17S U2 snRNP is an assembly intermediate and precedes bona fide 17S U2 snRNP assembly? Could the interaction with U2 snRNP result from DDX42 overexpression? Could the complex be a disassembly intermediate?

Response: We appreciate the reviewer for these queries. **First**, the interactions of DDX42 with SF3b and U2 snRNP should not be a result of

DDX42 overexpression, because these interactions were also identified using specific antibodies without overexpressing DDX42 (ref. 11: Will et al., *EMBO J.*, 2002; ref. 46: Zhao et al., *J. Biochem.*, 2022). **Second**, it's unlikely the DDX42-associated complexes are disassembly intermediates since DDX42 was not detected in any form of spliceosomes in the E-to-ILS splicing cycle. **Third**, DDX42 is associated with SF3b complex but is released upon formation of 17S U2 snRNP. Our structures reveal the anchoring site of DDX42 helicase domain on SF3B1 overlaps with that of U2 snRNA and TAT-SF1. The helicase domain has to be displaced from the C-terminus of SF3B1^{HEAT} upon recruitment of 12S U2 particle and TAT-SF1. Notably, DDX42 and DDX46 are co-localized in Cajal bodies, where the U2 snRNA is highly enriched (ref. 11: Will et al. *EMBO J.*, 2002). Through in vitro competitive assay, we show DDX46 could outcompete DDX42 from SF3b; the replacement of DDX42 by DDX46 may constitute a turning point in the maturation process of 17S U2 snRNP. Together, these structural and biochemical observations strongly suggest DDX42 is involved in the assembly of 17S U2 snRNP.

Can the authors explain how the data shown in Figure 1A indicates that both DDX42 and DDX46 can be stoichiometrically associated with SF3B1, especially when their binding is presumably mutually exclusive? The experimental method for this experiment is missing and should be included. The authors also use this data to conclude that DHX15 has weaker binding to U2 snRNP. Can they rule out the alternative explanation that DHX15 association with U2 snRNP is regulated, and only a subset of the U2 snRNP present in cells would be found in the conformation to which DHX15 binds?

Response: We appreciate the reviewer for these queries. In Fig.1a, the mass spectrometric analyses identified peptides derived from DDX42, DDX46 and DHX15 in immunoprecipitates prepared from HEK293F cells that expressed Flag-tagged SF3B1. Given SF3B1 is in over excess, it's possible that both DDX42 and DDX46 could be co-precipitated despite their binding to SF3B1 are presumably mutually exclusive. As requested by the reviewer, the experimental method is included in the method of revised manuscript. For the argument on DHX15, we totally agree with the reviewer's comment and would apologize for this over-interpretation. We have removed this statement in our revised manuscript.

The authors state that the so-called "loose" and "weak" interaction of the DDX42 helicase domain with the SF3b core is required by its activity as an RNA helicase for translocation. The authors should be careful when relating weak density to binding affinity, because the helicase domain may instead be interacting with one of the many flexible/disordered regions of the SF3b complex. Furthermore reference 20, which is cited to point out a so-called "RNA chaperone" activity for DDX42, provides evidence that DDX42 is an

RNA-dependent ATPase that does not translocate to unwind RNA, but instead uses ATP-hydrolysis to regulate binding and release of the protein from single stranded RNA. The protein can also disrupt RNA/protein interactions. Emphasizing that the target of DDX42 activity in the context of U2 snRNP assembly or function is unknown would be helpful for readers.

Response: Point accepted. We have carefully reviewed this section and rephrased the statement as the reviewer suggested. In the discussion section, we also claimed that the target of DDX42 is currently unclear.

The methods state that the cell line utilized is Hek293F, but results refer to them as Hek293. This should be rectified to match as they are different lines.

Response: Point accepted. The cell line used here is HEK293F. This is corrected in revised manuscript.

For figure 1D, labels of the different protein components would be helpful as the yellow from the EM map distorts the underlying colors. Alternatively, a gray map would

Response: Point accepted. The labels of several key components (TAT-SF1, DDX46 and U2 snRNA) were added. The remaining components, which remain unchanged from pre-17S-to-17S U2 snRNP, are left uncolored.

For Figure 3, “TATSF1” should be included in the labels for its different domains, which should have the same color.

Response: Point appreciated. Three domains of TAT-SF1 are colored differently because the UHM domain is missing in the structure of pre-17S U2 snRNP. To avoid misunderstandings, we explained the coloring and domain organization of TAT-SF1.

The manuscript has a number grammatical and spelling errors. A few of these are highlighted below:

Line 44- “spliceosome” should be preceded by an article. Typically, “the spliceosome” is used.

Line 119- DXH15 should be DHX15

Line 173- knowing to known

Response: We thank the reviewer for pointing out these errors, which have been corrected in the revised manuscript.

REVIEWERS' COMMENTS

Reviewer #1 (Remarks to the Author):

The authors have addressed all my concerns in the revised manuscript except for two minor points.

In Fig. 3a, I appreciate the authors' detailed explanation. My real question was whether the corresponding space occupied by DDX46 in the 17S U2 snRNP is not occupied by DDX42 in the pre-17S U2 snRNP as shown in the Figure.

In Fig. 5a, my question was why the RRM domain of the free form of TAT-SF1 is missing. I realize now that it is not missing but is hard to be distinguished from the Linker domain. I suggest the authors make the RRM domain more distinguishable.

Reviewer #2 (Remarks to the Author):

In revising their manuscript, the authors have adequately addressed all points raised by this reviewer.

Reviewer #3 (Remarks to the Author):

In the revised manuscript, Yang et al. adequately addressed most of my major concerns. I still suggest a few minor revisions to the text to qualify the interpretation of their findings more carefully, and to clear up some confusing passages.

1) I remain unconvinced that "17S U2 snRNP assembly precursor" or "pre-17S U2" is a fully qualified title for the DDX42-U2 snRNP. Instead, the authors should refer to this idea as a hypothesis. Indeed, the argument provided in their rebuttal could be added to the introduction or discussion to help readers understand the evidence supporting the hypothesis:

"Third, DDX42 is associated with SF3b complex but is released upon formation of 17S U2 snRNP. Our structures reveal the anchoring site of DDX42 helicase domain on SF3B1 overlaps with that of U2 snRNA and TATSF1. The helicase domain has to be displaced from the C-terminus of SF3B1HEAT upon recruitment of 12S U2 particle and TAT-SF1. Notably, DDX42 and DDX46 are co-localized in Cajal bodies, where the U2 snRNA is highly enriched (ref. 11: Will et al. EMBO J., 2002). Through in vitro competitive assay, we show DDX46 could outcompete DDX42 from SF3b; the replacement of DDX42 by DDX46 may constitute a turning point in the maturation process of 17S U2 snRNP. Together, these structural and biochemical observations strongly suggest DDX42 is involved in the assembly of 17S U2 snRNP."

2) In my opinion, their statement that DDX42 helicase activity is "likely" responsible for BSL formation is also an over-reach. Similarly, the statement at line 253 "The action of DDX 42 on U2 snRNA results in displacement of the helicase domain..." is too speculative. Other possible functions include remodeling of Stem IIA/C, preventing non-functional SF3B interactions with 12S U2 snRNP, etc. The relative position of a poorly resolved density is not sufficient to rule out these other possibilities.

3) Line 140: Explain "loosely" in biochemical/structural terms. For example- similar to the description in line 162-165: "the interface between its helicase domain and the SF3B core is characterized by weak EM density".

4) Line 60: A better term for this family of proteins is "RNA-dependent ATPase".

5) Line 222: "are should be" to "are"

6) The paragraph discussing SF3B1 cancer mutations and DDX42 and DDX46 contacts at line 264 is very difficult to follow. The information became much clear to me when presented in Extended Table 4. I suggest moving this table to the manuscript body.

Point-by-Point response to reviewers' comments

We would like to thank the reviewers for reviewing the revised manuscript and deeming it appropriate for publication. Addressing their comments helped us significantly improve the quality of our manuscript. Please see below our point-by-point response to each of the remaining concerns raised by the reviewers.

Reviewer #1:

The authors have addressed all my concerns in the revised manuscript except for two minor points.

In Fig. 3a, I appreciate the authors' detailed explanation. My real question was whether the corresponding space occupied by DDX46 in the 17S U2 snRNP is not occupied by DDX42 in the pre-17S U2 snRNP as shown in the Figure.

Response: We apologize for this confusion. We are not sure whether the anchoring site of **DDX46 helicase domain** in 17S U2 is occupied by DDX42, since the exact location of DDX42 helicase domain remains unclear. But the anchoring sites of **DDX46 N-termini** (except DDX46 helix α 1) in 17S U2 snRNP are indeed occupied by the N-termini of DDX42 in pre-17S U2 (see the structure overlay in Fig.4b). Also, the binding site of DDX42 helicase domain in DDX42-SF3b complex is occupied by U2 snRNA and TAT-SF1 in pre-17S U2 snRNP (see the structure overlay in Fig. 3b inset).

In Fig. 5a, my question was why the RRM domain of the free form of TAT-SF1 is missing. I realize now that it is not missing but is hard to be distinguished from the Linker domain. I suggest the authors make the RRM domain more distinguishable.

Response: Point accepted. This figure has been modified; the RRM domain should now be more distinguishable.

We thank this reviewer for his/her comments.

Reviewer #2:

In revising their manuscript, the authors have adequately addressed all points raised by this reviewer.

Response: We are happy to have been able to address the comments and concerns raised to the reviewer's satisfaction.

We thank this reviewer for his/her time.

Reviewer #3:

In the revised manuscript, Yang et al. adequately addressed most of my major concerns. I still suggest a few minor revisions to the text to qualify the interpretation of their findings more carefully, and to clear up some confusing passages.

Response: We are happy to have been able to adequately respond to the reviewer's questions and comments. Please see below our point-by-point response to the remaining concerns raised by the reviewer.

1) I remain unconvinced that “17S U2 snRNP assembly precursor” or “pre-17S U2” is a fully qualified title for the DDX42-U2 snRNP. Instead, the authors should refer to this idea as a hypothesis. Indeed, the argument provided in their rebuttal could be added to the introduction or discussion to help readers understand the evidence supporting the hypothesis:

“Third, DDX42 is associated with SF3b complex but is released upon formation of 17S U2 snRNP. Our structures reveal the anchoring site of DDX42 helicase domain on SF3B1 overlaps with that of U2 snRNA and TATSF1. The helicase domain has to be displaced from the C-terminus of SF3B1HEAT upon recruitment of 12S U2 particle and TAT-SF1. Notably, DDX42 and DDX46 are co-localized in Cajal bodies, where the U2 snRNA is highly enriched (ref. 11: Will et al. EMBO J., 2002). Through in vitro competitive assay, we show DDX46 could outcompete DDX42 from SF3b; the replacement of DDX42 by DDX46 may constitute a turning point in the maturation process of 17S U2 snRNP. Together, these structural and biochemical observations strongly suggest DDX42 is involved in the assembly of 17S U2 snRNP.”

Response: Point accepted. We agree with the reviewer that further studies are required to fully address the functional roles of DDX42. To avoid controversies, we used the term “DDX42-U2 complex” to replace the previous term “pre-17S U2 snRNP” and claimed that the “17S U2 snRNP assembly precursor” is our hypothesis. The introduction section and the figures were updated as well.

2) In my opinion, their statement that DDX42 helicase activity is “likely” responsible for BSL formation is also an over-reach. Similarly, the statement at line 253 “The action of DDX 42 on U2 snRNA results in displacement of the helicase domain...” is too speculative. Other possible functions include remodeling of Stem IIA/C, preventing non-functional SF3B interactions with 12S U2 snRNP, etc. The relative position of a poorly resolved density is not sufficient to rule out these other possibilities.

Response: As the reviewer pointed out, some of the statements on DDX42 in previous manuscript could be inappropriate, leading to controversies. During revision, we carefully checked our manuscript and modified corresponding

sections. For example, the statement “*DDX42 helicase activity is likely responsible for BSL formation*” was reworded as “*This analysis suggests a potential role of DDX42 in the formation of BSL*”. The statement “...perhaps through unleash its RNA helicase activity on U2 snRNA” was entirely removed. The speculation “*The action of DDX 42 on U2 snRNA results in displacement of the helicase domain...*” was reworded as “*Stable accommodation of BSL on SF3B1^{HEAT} results in displacement of the helicase domain...*”

3) Line 140: Explain “loosely” in biochemical/structural terms. For example- similar to the description in line 162-165:” the interface between its helicase domain and the SF3B core is characterized by weak EM density”.

Response: Point accept. We modified this description trying to be more specific.

4) Line 60: A better term for this family of proteins is “RNA-dependent ATPase”.

Response: Point accepted. The term “RNA-dependent ATPase” was used in revised manuscript.

5) Line 222: “are should be” to “are”

Response: Corrected in the revised manuscript.

6) The paragraph discussing SF3B1 cancer mutations and DDX42 and DDX46 contacts at line 264 is very difficult to follow. The information became much clear to me when presented in Extended Table 4. I suggest moving this table to the manuscript body.

Response: We appreciate this suggestion, but we feel the information embedded in this table overlaps with Fig. 5b and the quality of this table is also insufficient to be included in the manuscript body. In addition, the effects of SF3B1 cancer mutations are complicated, we need to be careful on this issue since we do not provide further evidence in this manuscript. We believe our following-up studies on DDX42 and DDX46 could provide a clearer view on cancer mutations.

We thank this reviewer for his/her comments.